# Analysis of the Risk Impact of Implementing Digital Innovations for Logistics Management

**Agnieszka Barczak** [1], **Izabela Dembińska** [2] **and Łukasz Marzantowicz** [3,*]

[1]  Department of System Analysis and Finance, West Pomeranian University of Technology Szczecin, 70-310 Szczecin, Poland; agnieszka-barczak@zut.edu.pl
[2]  Faculty of Engineering and Economics of Transport, Maritime University of Szczecin, 70-500 Szczecin, Poland; i.dembinska@am.szczecin.pl
[3]  Department of Logistics, SGH Warsaw School of Economics, 02-554 Warszawa, Poland
*  Correspondence: lukasz.marzantowicz@sgh.waw.pl; Tel.: +48-22-564-9326

**Abstract:** The emergence of digital technology is a paradigmatic historical change. As a process of transforming social engineering structures, digitization has had a ubiquitous impact on the organization of structures and business logic, as well as on economic principles and rules. The fertile ground for digital technology applications is logistics management, which manifests itself in the dynamic development of logistics 4.0. Increasingly, it is pointed out that digital technology has some distinct features that have fundamental implications for innovation. The aim of the present study is to determine the impact of the risk of implementing digital technologies for logistics management. The study was conducted using the standardized questionnaire interview method with representatives of the management of enterprises. The attempt was random. The sampling was made up of micro, small, medium, and large enterprises from the production and services sectors, having a logistics unit or a logistics division, located in the "Bisnode Poland" database. In total, 360 full interviews were carried out. For the study, we defined macro-environment, operational, functional, and microenvironment risks. The basic conclusion is that between each type of risk and the type of digital technologies used in the studied entities and their partners in the supply chain, there is a high and very high dependence in the case of three-dimensional printing (3D printing), artificial intelligence, blockchain, drones, augmented reality, and self-propelled vehicles.

**Keywords:** logistics management; digital technologies; innovation; risk

## 1. Introduction

The aim of the present study is to determine the impact of the risk of implementing digital innovations for logistics management. For the needs of the research, a typology of digital technologies was used, characterizing digital innovation as the effect of applying these technologies. Here, specific technologies include cloud computing, the internet of things, three-dimensional printing (3D printing), artificial intelligence, big data analytics, blockchain, automation, robotics, drones, machine learning, augmented reality, self-propelled vehicles, and digital Platforms. The main question was about the possibility of identifying the risks associated with the implementation of digital technology and the relationships that arise between the type of risk and digital technology from the perspective of its impact on the logistic management process. We asked the following: What kind of risk does the sphere affect and with what force? The sub-objective of the study was also designated as the demonstration of the diversified risk arising from the implementation of the digital technology and the effects of these risks in terms of logistical management.

This article is a research analysis. The first part of the article presents the scope of knowledge based on the literature review, which allowed answers to be found concerning the research questions posed. By analyzing the current state of knowledge in the area of the main research areas, digital technologies, logistics management, and related risks, research gaps have been identified, where the fulfillment of which is to be the research carried out in the article. The following describes the applied methodology of the conducted study, at the same time determining the research hypothesis, and then the analysis of the data obtained from the study was made. In the part containing the result, the focus was on the analysis of the test results, which made it possible to verify the hypothesis and mark the degree of implementation of the objective set in the study. Furthermore, a scientific discussion was conducted, drawing conclusions from the analysis and delineating a plane for further research, leading to a conclusion on the future of digital technologies in logistics management.

## 2. Literature Review

Being innovative is nowadays not only a challenge for enterprises, but also more and more a must. Innovation is imposed on the one hand by the conditions of competition, while on the other hand, enterprises, business partners, and customers expect it. Innovation, in the simplest sense, refers to the introduction of something new or a new idea, method, or device [1]. In economics, scientific conceptualizations of innovation can be found from the 1930s, in the works of J. A. Schumpeter [2], who defined innovation as a commercial or industrial application of something new. Here, this could mean a new product, process or production method, a new market or source of supply, a new form of trade, or a new business or financial organization. The goal of innovation is to increase the efficiency of the enterprise. First of all, innovations affect the increase of competitive advantage or its maintenance. Secondly, innovations affect the possibility of winning or expanding markets and increasing product quality, consequently shifting the demand curve for a company's product. Innovations can also shift the company's cost curve. This aspect, as the third way to achieve the goal of implementing innovation, is particularly important when the production level and market position are primarily determined by the level of production cost, distribution, and transaction cost. The fourth way is a favorable change in the investment capacity of the company and, consequently, the creation of new knowledge and know-how [3].

With the rapid development of digital technologies such as ubiquitous computing, digital convergence, Web 2.0, service-oriented architecture, cloud computing, and the open source revolution, an important and desirable aspect of enterprise innovation is the ability and skill to use digital innovation. The emergence of digital technology is a historical paradigm shift. It shapes organizations and markets. Bresnahan and Trajtenberg [4] classify digital technology as "general purpose technology", comparing it with, for example, a steam engine. While the steam engine enabled the mechanization of production processes at the end of the 19th century, technological innovation is associated with industrialization as a wider process of social transformation [5]. Unfortunately, it is not an easy task to clearly define digital innovations. From the point of view of economic practice, this may not be a significant problem, but from the perspective of theory this is important, because of the order in the classification of innovations or due to the typology of innovation strategies. It can be assumed that the starting point should be a general understanding of digitalization. The Swiss International Institute for Management Development (The Swiss IMD Institute) defines the phenomenon of digital transformation as "a radical organizational change that companies make using modern technologies to achieve greater business efficiency". According to Oracle, the most important distinguishing feature of this process is the simultaneous application of five phenomena that emerged with the advent of modern information technologies (IT technologies), namely, social media, mobile devices, the internet of things, cloud computing, and real-time analytical systems [6].

By digital innovation, we mean an innovation enabled by digital technologies that leads to the creation of new forms of digitalization. By digitalization, we mean the transformation of socio-technical structures that were previously mediated by non-digital artifacts or relationships into ones that are

mediated by digitized artifacts and relationships. Digitalization goes beyond a mere technical process of encoding diverse types of analog information in digital format (i.e., "digitization") and involves organizing new socio-technical structures with digitized artifacts, as well as changes in artifacts themselves [7]. Other definitions of digital innovation interpret it as a technological change in the product. More specifically, about the use of digital technology in the product, which according to O. Henfridsson [8] has not been applied so far. On the other hand, digital innovation can be understood in the way presented by M. Akesson [9], treating digital innovation as a solution stimulated by digital technologies, but concerning new products and services. Theoretical mixed models in the scope of the definition of digital innovation do not distinguish the moment of application of digital technology, but they give it a mandatory value, that is decisive for digital innovation [3,10]. Assuming the above definitional assumptions, it is unquestionable to recognize that digital businesses create digital innovations understood from the perspective of the use of digital technologies. The use of digital technologies as a tool is not new. However, it is worth noting that digital technology alone is not enough to determine digital business. According to a study conducted by the World Economic Forum in 14 business sectors, it was pointed out that only in the case of appropriate joint applications of different technologies that there is an increase in return of investment in these technologies [11]. The diffusional use of technology and its complementation is what drives the emergence of digital business [12]. The causal relationship is clearly visible, stating that the source of innovativeness of enterprises is digital technologies, which shapes the perception of digital business in various sectors.

The enterprise is increasingly dependent on the digitization of the economy. According to various market forecasts, companies will have to face new challenges resulting from the digitization of the economy in the near future. It is necessary to reorganize management so that the company can easily and quickly absorb emerging digital technologies. Enterprises must find a way for more dynamic external and internal volatility and take into account the fact that not only the company is digitized, but also the whole of its environment becomes digital [13], creating new requirements for the business activity. There is also the point of view that innovations in an enterprise create a kind of ecosystem, building multilateral value chains [14]. This is also confirmed by John Hagel III and Marc Singer [15], who point to innovation as one of the new and obligatory areas of business activity. However, businesses use digital innovations in a breakthrough or incremental (radical) aspect. As C. Christensen points out [16], this distinction is necessary because breakthrough innovations guarantee differentiation in the market by proposing a completely new offer, while incremental innovations change the direction of thinking and allow you to dynamically control competitiveness. Both cases are reflected in one of the initial definitions introduced by J.A. Schumpeter, saying that innovation is also a new combination or configuration of existing factors or components of a product or service [2].

The use of modern digital technologies, i.e., building value chains based on the new generation of an operational model that creates an integrated management system, determines the perception of digital innovation from the perspective of management methods and tools. This correct statement of A. Bollard et al. [17] indicates a generational change in management, with respect to the need to build management systems based on the use of digital technologies. In literature, there is a reference to shaping innovation through knowledge, technology, and integration. Effective business management is determined by the ability to develop innovation [18], and management of innovation is today treated as a management concept [19,20].

Digital innovations and digital technologies stimulate mobility in terms of access to information and its analysis. They are the tools controlling logistics activities [21], which in reference to the concept of management raises the legitimate approach to digital innovation in logistics management. The literature presents many approaches regarding the relationship between digital innovation and logistics management. The essence of the problem is mainly the perception of digitization as a management tool. Digital innovations create digital supply chains and create hybrid systems of digital production as part of the logistics chain and conventional logistics [22]. Many views presented in the literature reflect the process approach to understanding logistics from the perspective of using digital

innovations. The inclusion of digital technologies in logistics processes is called logistical innovation [23]. Table 1 presents examples of solutions that imply the development of digital technologies in the area of logistics.

**Table 1.** Selected solutions, resulting from the implementation of digital technology in the area of logistics.

| Area | Key Solutions |
| --- | --- |
| Transport | Safety systems, route planning systems, unmanned trucks, intelligent transport management systems, intelligent highway, navigation systems, augmented reality |
| Warehouse management | Radio-Frequency Identification (RFID), intelligent warehousing, intelligent distribution center, intelligent forklift, intelligent racking, automation of picking, augmented reality |
| Production | Production control systems, quality control systems, intelligent assembly systems |
| Supply chain | E-supply chain, e-commerce, virtual supply network |

Source: Own elaboration.

The implementation of digital technologies in the logistics area has led to the emergence of so-called intelligent logistics (also known as smart logistics) [24]. M. Weiser [25] notes that the term "intelligent" expresses current technological changes, which shows its dependence on time. Therefore, he thinks that one can agree that the word "intelligent" should be understood in this context in the context of the implementation of innovation and the availability of state-of-the-art technology. Sah, in turn, thinks that everything that limits human efforts and automates tasks should be described as "intelligent" [26].

According to a PwC survey, industry experts report that enthusiasm in the transport and logistics industry (T and L) regarding the adoption of new digital technology surpasses that of any other industry, where the T and L industry scores 90% as opposed to 83% in other industries. However, the survey also opines that the lack of digital culture is by far the biggest hurdle that the industry faces, as evident from the illustration below. Transportation infrastructure and streamlined business processes are key factors that determine the growth of the T and L industry. Digital technology is a boost to the infrastructure of industries, and logistics industry is no different. Improved infrastructure, in turn, positively contributes to flexible and scalable logistics processes [27,28].

Looking for studies on the relationship between the risk of implementing digital technology and the area of logistics management, we can point to studies loosely related to this problem. D. Ivanov, A. Dolgui, and B. Sokolov have analyzed the impact of digital technologies on risk in logistics management [29]. The problem of risk is also presented in the aspect of cyber security [30].

The literature review allows the claim that the problem of the risk of implementing digital technology in the area of logistics management is not yet well recognized. As a rule, research is being talked about the areas of implementing digital technology, digitally derived strategies or the efficiency of using digital technologies. Therefore, it can be concluded that research into the risk of implementing digital technology in the area of logistics management is a research gap both in the area of management and in the narrower perspective, namely, in logistics. Therefore, the research presented in the further part of the study is justified and will fill this gap to some extent.

## 3. Materials and Methods

### 3.1. Research Hypothesis

The following research hypotheses were adopted for the purposes of the main research objective:

**Hypothesis 1 (H1).** *The risk of introducing digital technologies in the enterprise determines the impact on logistics management.*

**Hypothesis 2 (H2).** *Digital technologies (as determinants of digital innovations), depending on the type, have a diversified impact on the management of logistics processes.*

*3.2. Method of Carrying out the Research*

The study was conducted using standardized questionnaire interviews, i.e., interviews containing questions with a strictly defined order and unchanged wording, where the questions are generally closed. The standardized questionnaire interviews are derived from the neo-positivist research paradigm. The neo-positivist research paradigm should be understood as a functionalist paradigm. In other words, it requires a systemic approach, oriented to creating systems and verifying the truth using quantitative methods that assume the possibility of generalization and mathematical modeling. In this approach, the applied methodology is objective and universal [31]. Although, with the questionnaire interviews, their role was also played by the interpretive paradigm. The interpretive paradigm should be understood as an explanation of a phenomenon, but not a suggestion. The thinking-minded aspect is noticeable. It is used if, apart from quantitative methods, the research is also qualitative [31]. The critical postmodern paradigm was also used. Here, the paradigm assumes the existence of an objective social reality that requires reconstruction. This paradigm adopts a critical attitude to the achievements of social sciences. With this paradigm, researchers should strive to expose false traps of collective consciousness [31]. As part of this research paradigm, one may strive to discover the truth about the world in a systematic, standardized, factual, synthesizing, non-subjective, and cumulative way [31]. This is a well-established research method. Its historical origins are combined with research conducted by Arthur Bowley and William Benett-Hurst in Great Britain in 1912, regarding the living conditions of the working class in the cities of Stanley and Reading. However, the most historically significant contribution is that of George Gallup, who in 1940, in the 1940 population census, carried out research on a five-percent sample of the American population [32].

Technological modification of the above-mentioned method was used, in which the direct contact of the interviewer with the respondent was abandoned in favor of communication by means of a telephone, and the traditional paper questionnaire was replaced by a computer. This technique is known as computer-assisted telephone interviewing (CATI). In relation to classic interviews, telephone interviews with computer support have a number of methodological characteristics that make them particularly useful in the present study. First of all, CATI is a technique with a very high level of standardization, allowing the introduction of only predefined data in terms of their form and content. Secondly, telephone surveys enable constant control over gathering data and the continuous monitoring of sample size and respondents' responses. Thirdly, the CATI technique provides the opportunity to conduct research in the representative sector of the company due to the availability of the entire sampling frame (REGON database), elimination of so-called clustering (the focus of entities in geographically close points), and the enablement of the draw procedure, thanks to the electronic form of this database. In addition, CATI surveys allow combination online surveys and other quality techniques, including projection tests. CATI is a technique that requires significantly smaller financial and organizational expenses than in the case of the classic face to face (F2F) questionnaire survey method. The CATI technique makes it possible to modify the research tool even after the start of the field test phase, where questions and even blocks of questions can then be added or modified. The most important advantage, and at the same time, the characteristics of this research method, is the fact that on the basis of a properly selected research sample, it is possible to generalize the results into a population.

*3.3. Selection of the Sample in the Study and the Methodology of the Research Analysis*

The sample was random. The study was conducted with representatives of the company's management staff. The sampling was made up of enterprises from the production and services sectors, having a logistics unit or a logistics division, located in the Bisnode Poland database. In total, 4237 companies were contacted, and 360 full interviews were carried out. The randomization algorithm

embedded in the telephone research software provided the same chance of being in the test for each of the records in the database. Despite the fact that the sample was statistically correct, we analyzed data from 120 companies and not from 360.

The rationale for the selection of this sample size is the analysis carried out. There are many formulas for calculating the sample size, depending on the available information [33–35]. The following relationship was selected for the analysis [36]:

$$n = \frac{p(100 - p)z^2}{e^2}.$$

where $n$ is the required sample size, $p$ is the percentage occurrence of a state or condition, $e$ is the percentage maximum error required, and $z$ is the value corresponding to level of confidence required.

For these studies, $p = 50\%$ and $e = 9\%$ were accepted. The result obtained (118.5677) has been rounded up to full tens.

Because in economic research it is often impossible to examine the whole population, after establishing a representative sample, the methods of the so-called statistical inference have been used. They allow the generalization of the results obtained from the sample to the entire population.

The confidence interval for the structure index allows the analysis of the share of individual characteristics in relation to the whole population, based on data from the sample [37–40]. The confidence interval for the structure index was determined from a large sample (independent sampling) of a particular variant of the analyzed research feature from the entire statistical population. Here, this was based on the following formula:

$$\frac{m}{n} - u_\alpha \sqrt{\frac{\frac{m}{n}\left(1 - \frac{m}{n}\right)}{n}} < p < \frac{m}{n} + u_\alpha \sqrt{\frac{\frac{m}{n}\left(1 - \frac{m}{n}\right)}{n}}$$

where $m$ is the number of elements highlighted in the sample and $n$ is the sample size.

The value of $u_\alpha$ statistics is read from a table of the cumulative distribution function for a standard normal random variable for $1 - \frac{\alpha}{2}$.

Because the research sample was selected correctly, the results indicated in brackets (confidence intervals determined for the significance level of $1 - \alpha = 0.95$) can be generalized to all enterprises in the Bisnode Poland database.

"Correlation may be described as the degree of association between two variables ( … ). In general, we can say that the study of interdependence leads to the investigation of correlations" [41]. "Correlation analysis is a term used to denote the association or relationship between two (or more) quantitative variables. This analysis is fundamentally based on the assumption of a straight line [linear] relationship between the quantitative variables. Similar to the measures of association for binary variables, it measures the "strength" or the "extent" of an association between the variables and also its direction" [42].

The study also used Spearman's rank correlation coefficient, which has been defined as the Pearson correlation coefficient calculated for the variable ranks [43] (the rank is the number that corresponds to the place in the order of each feature). The difference also applies to the fact that it measures any monotonous dependence [44], including if the features are qualitative. It is designated by the following formula [45]:

$$r = 1 - \frac{6 \sum_{i=1}^{n} d_i^2}{n(n^2 - 1)}$$

where $d_i^2$. is the difference between the ranks of the corresponding variable features.

The test of significance for Spearman's rank-order correlation coefficient is a test used to verify the hypothesis about the monotonic relationship between the studied features of the population. It is

based on the Spearman rank correlation coefficient. The following hypotheses were formulated from the test:

Main hypothesis: $H_0$: $r = 0$, saying that the features are not correlated (statistically significant),
Alternative hypothesis: $H_1$: $r \neq 0$, saying that there is a correlation between features (statistically insignificant).

The test statistics have the form:

$$p = \frac{r}{\sqrt{\frac{1-r^2}{n-2}}}$$

The statistics have a Student's t distribution at $n - 2$ degrees of freedom. Comparing the results of the statistics obtained with the assumed level of significance ($\alpha = 0.05$), one should decide whether to accept or reject the main hypothesis. The hypothesis $H_0$ should be rejected in favor of the alternative hypothesis (statistically significant) if $p \leq \alpha$. However, if $p > \alpha$ there are no grounds to reject the $H_0$ hypothesis (statistically insignificant).

The correlation coefficients determined were used to calculate the determination coefficient, based on the formula:

$$R = r^2 \cdot 100\%$$

Thanks to this, it was possible to determine the impact of one feature on another, in percentage terms. When interpreting the results, the classification according to J. Guilford was used, where:

$|r| = 0$—lack of correlation
$0.0 < |r| \leq 0.1$—dim correlation
$0.1 < |r| \leq 0.3$—weak correlation
$0.3 < |r| \leq 0.5$—average correlation
$0.5 < |r| \leq 0.7$—high correlation
$0.7 < |r| \leq 0.9$—very high correlation
$0.9 < |r| < 1.0$—almost full correlation
$|r| = 1$—full correlation.

This classification is suitable for use in the analysis of Spearman's rank correlation coefficients. The purpose of the introduction to further analysis was to briefly characterize the subjects studied. Table 2 presents the share of enterprises, in terms of the sector of activity, in the research sample and the confidence interval for the structure indicator. The table also presents the breakdown of enterprises due to the geographical scope of operations.

**Table 2.** Division of the surveyed entities.

| Enterprise | Participation in the Sample | Confidence Interval |
|---|---|---|
| Production | 40.83% | $36.34\% < p < 45.32\%$ |
| Service | 54.17% | $45.26\% < p < 63.08\%$ |
| Production and service | 5.00% | $1.10\% < p < 8.90\%$ |
| Regional | 39.17% | $30.44\% < p < 47.90\%$ |
| Poland | 92.50% | $87.79\% < p < 97.21\%$ |
| European Union | 71.17% | $66.34\% < p < 82.00\%$ |
| Europe | 55.00% | $46.10\% < p < 63.90\%$ |
| North America | 19.17% | $12.13\% < p < 26.21\%$ |
| South America | 20.83% | $13.56\% < p < 28.10\%$ |
| Asia | 22.50% | $15.03\% < p < 29.97\%$ |
| Africa | 10.00% | $4.63\% < p < 15.37\%$ |
| Australia | 7.50% | $2.79\% < p < 12.21\%$ |

Source: Own study, based on the results of the questionnaire surveys.

By limiting the activity to the regional market, 10% of production enterprises ($4.63\% < p < 15\%$) and 29.17% of service enterprises operate in the region ($21.04\% < p < 37.30$). The whole of Poland is covered by 53.33% of the service enterprises ($44.40\% < p < 62.26\%$), 25.68% of the production companies ($25.68\% < p < 42.66\%$) and 5.00% of the production and service entities ($1.10\% < p < 8.90\%$). Overall, 39.17% of service companies ($30.44\% < p < 47.90\%$), 30.83% of production ($22.57\% < p < 39.09\%$) and 4.17% of production and service companies ($0.59\% < p < 7.75\%$) operate in countries in the European Union. The economic activity on the entire European continent is conducted by 26.67% of service enterprises ($18.76\% < p < 34.58\%$), 25.83% of production enterprises ($17.99\% < p < 33.66\%$) and by 2.50% of production and service companies ($0.00\% < p < 5.29\%$). Overall, 10.83% of production enterprises and 8.33% of service enterprises ($6.22\% < p < 10.44\%$) cooperate with North American countries ($5.27\% < p < 16.39\%$). With South American countries, 11.67% of production enterprises ($5.96\% < p < 17.4\%$), 8.33% of service enterprises ($6.22\% < p < 10.44\%$) and 0.83% of production and service enterprises ($0.00\% < p < 2.45\%$) cooperate there. A small part of the surveyed entities cooperate with partners from Asia, Africa, and Australia. The cooperation with Asian countries is carried out by 15.00% of the production enterprises ($8.61\% < p < 21.39\%$) and 7.50% of the service enterprises ($2.79\% < p < 12.21\%$). Here, 7.50% of production companies contacted African countries ($2.79\% < p < 12.21\%$) and 2.50% of service enterprises ($0.00\% < p < 5.29\%$). In the Australian markets, the values were 5.83% for production companies ($1.64\% < p < 10.02\%$) and 1.67% for service entities ($0.00\% < p < 3.96\%$). The analysis of interdependencies between the sector of activity and its geographical scope indicates a small dependence that is statistically insignificant (p) ($r = 0.1353$, $p = 0.1406$, $R = 1.83\%$).

## 4. Analysis in the Area of Risk Identification Related to the Implementation of Digital Innovations

### 4.1. Analysis

Taking into account the issue of risk types, which are identified by entities as dominant in the case of implementation and use of digital innovation, confidence intervals for the structure indicator were determined (Table 3). Based on the obtained results, it can be concluded that with a probability of 0.95 that the designated intervals contain an unknown share of all enterprises identifying with the indicated types of risk. The largest group of surveyed enterprises indicated that the risk of macroenvironment was dominant in the implementation and use of digital innovation.

**Table 3.** Evaluation of risk in the entities.

| Types of Risk | Participation in the Sample | Confidence Interval |
|---|---|---|
| Risk of macroenvironment | 56.67% | $47.80\% < p < 65.54\%$ |
| Operational risk | 17.50% | $10.70\% < p < 24.30\%$ |
| Functional risk | 16.67% | $10.00\% < p < 23.34\%$ |
| Risk of microenvironment | 9.17% | $4.01\% < p < 14.33\%$ |

Source: Own study, based on the results of the questionnaire surveys.

As mentioned in the section describing the research methodology, this consisted of choosing a predefined answer. As indicated in Table 3, the following risks were selected: Macroenvironment risk, operational risk, functional risk, and microenvironment risk. The risk of the macroenvironment should be defined as a source of external variables that imply the possibility of failure or opportunities that a company can achieve as a result of the use of digital technologies, taking into account market and partner relations. Operational risk is perceived as a set of variables in the area of process management. Functional risk is defined as business limitations at the interface between all functional departments of the company, and the risk of microenvironment concerns, in particular, decisions and relationships within the organization.

Interesting conclusions can be provided by the analysis of correlation coefficients by means of which the strength of dependence has been surveyed between:

- The types of risk identified as dominant in the implementation and use of digital innovation and the business sector, where the analysis unambiguously indicates a moderate correlation between the analyzed features and the correlation coefficient is statistically significant ($r = 0.3207$, $p = 0.0004$, $R = 10.28\%$).
- The risk identified as dominant in the implementation and use of digital innovation and the area of doing business, where in the case of the aggregation of area-related variables, the correlation coefficient is 0.1357 ($p = 0.1357$, $R = 1.84\%$), which indicates a very small correlation (statistically insignificant). Therefore, in Table 4, these values are presented, omitting the aggregation process.

**Table 4.** Correlation analysis.

| Area of Activity | Correlation Coefficient | Significance Level ($p$) | Coefficient of Determination |
|---|---|---|---|
| Regional | 0.5097 | <0.0001 | 25.98% |
| Poland | 0.5964 | <0.0001 | 35.57% |
| European Union | 0.2911 | 0.0013 | 8.47% |
| Europe | 0.1566 | 0.0876 | 2.45% |
| North America | 0.3359 | 0.0002 | 11.28% |
| South America | 0.2849 | 0.0016 | 8.12% |
| Asia | 0.2338 | 0.0102 | 5.47% |
| Africa | 0.4023 | <0.0001 | 16.18% |
| Australia | 0.4234 | <0.0001 | 17.93% |

Source: Own study, based on the results of the questionnaire surveys.

At the significance level $\alpha = 0.05$, it can be concluded that almost all the correlation coefficients presented in Table 4 are statistically significant. Here, an exception is the correlation coefficient for companies operating in Europe. As the obtained results indicate, the strength of interdependence is the highest in the case of enterprises operating in Poland and also on a regional scale. However, as indicated by the values of the determination index, this impact is small.

*4.2. Result of the Analysis*

Each economic operator should strive to minimize the risk incurred. The conducted research indicates that only 6.67% (2.21% < $p$ < 11.13%) of the analyzed enterprises have a dedicated cell in their structure responsible for risk management related to the implementation and use of digital innovation. However, it should also be noted that enterprises that do not have a separate department dealing with risk management have and use tools to prevent or limit the impact of unpredictable factors on the level of effectiveness of implementation and use of digital innovation. This group includes 41.67% (32.85% < $p$ < 50.49%) of enterprises. It is equally important to measure the impact of risk on the effectiveness of implementation and subsequent use of digital innovation. Such analysis is only performed by 29.17% (21.04% < $p$ < 37.30%) of the surveyed entities.

The change in the management of logistic processes resulting from the application of digital innovations has a large impact on the organization of the supply chain. In the analyzed entities, in 51.67% (42.73% < $p$ < 60.61%) of cases, the introduced modifications affected a change in the approach to cooperation with suppliers and customer relations. Here, 30.00% (21.80% < $p$ < 38.20%) of entities changed their approach in terms of improving the management within the enterprise, and 6.67% (2.21% < $p$ < 11.13%) of companies reconfigured the supply chain with the involvement of each chain made. Interestingly, 11.67% (5.93% < $p$ < 17.4%) of enterprises did not notice changes in the area of logistics and the supply chain. As the analysis shows, both entities in the service sector dominate in the group as well as in the entire population of enterprises. For the most part, the activity was carried out in Poland, the European Union, and as well as in other European countries. A small number of entities decide to cooperate with African and Australian contractors. It is worth noting, however, that the study of correlation and determination coefficients indicates a weak or moderate relationship between variables here. This allows us to conclude that the business sector and the geographic area of its operation have a small impact on the perception of risk in business.

## 5. Results in the Field of the Impact of Digital Technology on Logistics Management

For each of the defined types of risk, correlation relationships with the type of digital technology were determined. Using the correlation coefficient described in the previous subsection, it was possible to determine the strength of this correlation, which results in the possibility of adopting the perspective of assessing this dependence from the point of view of digital technology, but not risk. The adoption of such a point of view requires checking all identified risks, and not just those indicated as dominant. This is necessary to determine dominant digital technologies. The strength of correlation between risk types and digital technologies is presented in Table 5.

**Table 5.** Strength of correlation between digital technology and the type of risk.

| Types of Risk Identified as Dominant in the Implementation and Use of Digital Innovation | Digital Technologies Used in the Surveyed Companies and Their Partners in the Supply Chain | Correlation Coefficient | Significance Level ($p$) | Strength of Correlation [1] |
|---|---|---|---|---|
| Macroenvironment risk | Cloud computing | 0.2995 | 0.0009 | Weak |
| | Internet of things | 0.3814 | <0.0001 | Average |
| | 3D printing | 0.5671 | <0.0001 | High |
| | Artificial intelligence | 0.6354 | <0.0001 | High |
| | Big data analytics | 0.2346 | 0.0099 | Weak |
| | Blockchain | 0.6316 | <0.0001 | High |
| | Automation | 0.3297 | 0.0002 | Average |
| | Robotics | 0.3526 | <0.0001 | Average |
| | Drones | 0.6316 | <0.0001 | High |
| | Machine learning | −0.1461 | 0.1113 | Weak |
| | Augmented reality | 0.6316 | <0.0001 | High |
| | Self-propelled vehicles | 0.6084 | <0.0001 | High |
| | Digital platforms | 0.3729 | <0.0001 | Average |
| Operational risk | Cloud computing | 0.4663 | <0.0001 | Average |
| | Internet of things | 0.5842 | <0.0001 | High |
| | 3D printing | 0.7264 | <0.0001 | Very high |
| | Artificial intelligence | 0.8001 | <0.0001 | Very high |
| | Big data analytics | 0.4930 | <0.0001 | Average |
| | Blockchain | 0.7834 | <0.0001 | Very high |
| | Automation | 0.4350 | <0.0001 | Average |
| | Robotics | 0.4130 | <0.0001 | Average |
| | Drones | 0.7834 | <0.0001 | Very high |
| | Machine learning | 0.0206 | 0.8233 | Dim |
| | Augmented reality | 0.7834 | <0.0001 | Very high |
| | Self-propelled vehicles | 0.7667 | <0.0001 | Very high |
| | Digital platforms | 0.4926 | <0.0001 | Average |
| Functional risk | Cloud computing | 0.4452 | <0.0001 | Average |
| | Internet of things | 0.4810 | <0.0001 | Average |
| | 3D printing | 0.7109 | <0.0001 | Very high |
| | Artificial intelligence | 0.7587 | <0.0001 | Very high |
| | Big data analytics | 0.4499 | <0.0001 | Average |
| | Blockchain | 0.7917 | <0.0001 | Very high |
| | Automation | 0.3880 | <0.0001 | Average |
| | Robotics | 0.5406 | <0.0001 | High |
| | Drones | 0.7917 | <0.0001 | Very high |
| | Machine learning | 0.0296 | 0.7483 | Dim |
| | Augmented reality | 0.7917 | <0.0001 | Very high |
| | Self-propelled vehicles | 0.7751 | <0.0001 | Very high |
| | Digital platforms | 0.5134 | <0.0001 | High |
| Microenvironment risk | Cloud computing | 0.5111 | <0.0001 | High |
| | Internet of things | 0.4967 | <0.0001 | Average |
| | 3D printing | 0.8285 | <0.0001 | Very high |
| | Artificial intelligence | 0.8459 | <0.0001 | Very high |
| | Big data analytics | 0.5609 | <0.0001 | High |
| | Blockchain | 0.8751 | <0.0001 | Very high |
| | Automation | 0.4341 | <0.0001 | Average |
| | Robotics | 0.5538 | <0.0001 | High |
| | Drones | 0.8751 | <0.0001 | Very high |
| | Machine learning | 0.1287 | 0.1612 | Weak |
| | Augmented reality | 0.8751 | <0.0001 | Very high |
| | Self-propelled vehicles | 0.8604 | <0.0001 | Very high |
| | Digital platforms | 0.6714 | <0.0001 | High |

Source: Own study, based on the results of the questionnaire surveys. [1] Classification according to J. Guilford.

At the significance level $\alpha = 0.05$, it can be concluded that almost all the correlation coefficients presented in Table 5 are statistically significant. Here, the exception is the correlation coefficients for machine learning for each type of risk. The analysis of correlation coefficients between risk types identified as dominant in the case of the implementation and use of digital innovation and the type of digital technologies used in the surveyed entities and their partners in the supply chain show a high and very high dependence, observed with each type of risk in the case of 3D printing, artificial intelligence, blockchain, drones, augmented reality, and self-propelled vehicles.

The above statements imply the conclusion that both for the risk of macroenvironment indicated as dominant, and for other types of risk, the limited scope of the impact of technology was determined, showing the high and very high correlation strength of the selected technologies.

By means of the same method, the influence of the dominant risk on management was assessed. The impact of risk on management, as shown in Table 6, was determined based on the correlation strength.

At the significance level $\alpha = 0.05$, it can be concluded that all correlation coefficients presented in Table 6 are statistically significant. The values of correlation coefficients between risk types that have been identified as dominant in the implementation and use of digital innovation and elements affected by the applied digital innovation are presented in Table 5. The high and very high dependence observed with each type of risk occurs in the case of building the competitive advantage of the company in the next 3–5 years and short-term changes, due to the fact that the life cycle and suitability of digital innovation is too short to build long-term partner relationships using said technologies.

**Table 6.** Effect of implementation of digital technology on management.

| Types of Risk Identified as Dominant in the Implementation and Use of Digital Innovation | The Applied Digital Innovation Will Affect | Correlation Coefficient | Significance Level (*p*) | Strength of Correlation |
|---|---|---|---|---|
| Macroenvironment risk | Building a competitive advantage | 0.5012 | <0.0001 | High |
| | Increasing employment | 0.3834 | <0.0001 | Average |
| | Increasing market share | 0.3344 | 0.0002 | Average |
| | Starting operations in new markets | 0.3817 | <0.0001 | Average |
| | Building a competitive advantage in a strategic way | 0.3382 | 0.0002 | Average |
| | Support during strategy building and will not be of key importance | 0.336 | 0.0002 | Average |
| | Operational activities of the company | 0.4713 | <0.0001 | Average |
| | Strategy for building partner relations | 0.2568 | 0.0046 | Weak |
| | Support during building partner relations and will not have a key meaning | 0.2609 | 0.0040 | Weak |
| | Short-term changes, because the life cycle and suitability of digital innovation is too short to build long-term partner relationships using technology in the company | 0.571 | <0.0001 | High |
| Operational risk | Building a competitive advantage | 0.7046 | <0.0001 | Very high |
| | Increasing employment | 0.4614 | <0.0001 | Average |
| | Increasing market share | 0.5474 | <0.0001 | High |
| | Starting operations in new markets | 0.5008 | <0.0001 | High |
| | Building a competitive advantage in a strategic way | 0.5468 | <0.0001 | High |
| | Support during strategy building and will not be of key importance | 0.5501 | <0.0001 | High |
| | Operational activities of the company | 0.6087 | <0.0001 | High |
| | Strategy for building partner relations | 0.4674 | <0.0001 | Average |
| | Support during building partner relations and will not have a key meaning | 0.4766 | <0.0001 | Average |
| | Short-term changes, because the life cycle and suitability of digital innovation is too short to build long-term partner relationships using technology in the company | 0.7367 | <0.0001 | Very high |

**Table 6.** *Cont.*

| Types of Risk Identified as Dominant in the Implementation and Use of Digital Innovation | The Applied Digital Innovation Will Affect | Correlation Coefficient | Significance Level (*p*) | Strength of Correlation |
|---|---|---|---|---|
| Functional risk | Building a competitive advantage | 0.6649 | <0.0001 | High |
| | Increasing employment | 0.6201 | <0.0001 | High |
| | increasing market share | 0.509 | <0.0001 | High |
| | Starting operations in new markets | 0.5507 | <0.0001 | High |
| | Building a competitive advantage in a strategic way | 0.4661 | <0.0001 | Average |
| | Support during strategy building and will not be of key importance | 0.4643 | <0.0001 | Average |
| | Operational activities of the company | 0.6415 | <0.0001 | High |
| | Strategy for building partner relations | 0.3892 | <0.0001 | Average |
| | Support during building partner relations and will not have a key meaning | 0.3926 | <0.0001 | Average |
| | Short-term changes, because the life cycle and suitability of digital innovation is too short to build long-term partner relationships using technology in the company | 0.6954 | <0.0001 | High |
| Microenvironment risk | Building a competitive advantage | 0.7634 | <0.0001 | Very high |
| | Increasing employment | 0.6073 | <0.0001 | High |
| | Increasing market share | 0.6595 | <0.0001 | High |
| | Starting operations in new markets | 0.5614 | <0.0001 | High |
| | Building a competitive advantage in a strategic way | 0.6435 | <0.0001 | High |
| | Support during strategy building and will not be of key importance | 0.6442 | <0.0001 | High |
| | Operational activities of the company | 0.7493 | <0.0001 | Very high |
| | strategy for building partner relations | 0.5655 | <0.0001 | High |
| | Support during building partner relations and will not have a key meaning | 0.5704 | <0.0001 | High |
| | Short-term changes, because the life cycle and suitability of digital innovation is too short to build long-term partner relationships using technology in the company | 0.8149 | <0.0001 | Very high |

Source: Own study, based on the results of the questionnaire surveys.

## 6. Discussion

The analysis of the conducted study ambiguously indicates the pace of implementation of digital technologies. There is still a question about the innovativeness of companies in a situation where the impact of using digital innovations is basically a regional one. A general conclusion can be made that the implementation and use of digital technologies leads to an increase in the enterprise's innovation level, but at the same time, implies changes in logistic management in a way that will build or gain a competitive advantage in the next 3 to 5 years. It should be noted that the greatest impact on changes in logistic management will have, in the opinion of the respondents, only some technologies, bringing with them not fully-verified effects of the risk of implementing digital technologies.

The predefined types of dominant risk related to the implementation of digital technology and the determination of the impact of risk related to the implementation of digital technology on logistic management, defined by the expected effect, realizes the objective of the study set in the article. Nevertheless, one should pay attention to a certain degree of universality of the indicated effects. In general, the study showed that as many as 11.67% of the surveyed enterprises do not notice or do not foresee any changes in the area of logistics related to the implementation and use of digital technology. This is surprising, because at the same time, despite the recognition that changes may be slight, due to the fact that digital technology has too short a life cycle, at the same time, this indicates another extreme effect, namely, causing the ability to build or maintain a competitive advantage in the next 5 years to be impacted.

The main hypothesis has been positively verified. In terms of changes in logistics management, the study showed that variables based on the risk of implementing digital innovations can bring the following effects:

1. Building a company's competitive edge in the next five years shows a high dependence on variables that define the risk of macroenvironment and a very high dependency in the case of variables defining operational, functional, and micro-environment risk. It should be pointed out that the strength of dependence is highest in the types of operational, functional and microenvironment risk, but the scale of the phenomenon is less dependent on the size and scope of the company's operation. As the study showed, the largest number of enterprises indicated the risk of macroenvironment as the dominant one in the case of the implementation of digital technologies.

2. Due to the short lifecycle and usefulness of digital innovation, short-term management changes should be expected. A high dependence of the functional risk and macroenvironment risk variables was demonstrated, along with a very high dependence in the cases of operational and microenvironment risk. It should be pointed out that a larger number of enterprises indicated the macroenvironment risk of implementing digital technologies as dominant. The research analysis showed the greatest dependence in the case of operational and microenvironment risk, which is related to the diversity of technologies implemented in enterprises. The whole set of different technologies was tested, therefore, it is impossible to show the influence of one of them. Nevertheless, the result of this part of the analysis should be accepted as credible.

The above, seemingly extreme, and maybe even contradictory effects, however, are justifiable, because due to the wide collection of different technologies, enterprises only referred to the effects of the risk of implementing given technologies, previously pointing to the strength of the relationship between the type of risk and digital technology. Therefore, it is important to verify the auxiliary hypothesis. According to the analysis of the study, after indicating the dominant risk, the digital technologies that have the greatest impact on management and at the same time constitute the largest group of variables that create risk are 3D printing, artificial intelligence, blockchain, drones, augmented reality, and self-propelled vehicles. The high or very high dependence of technology has been indicated in relation to each identified type of risk. At the same time, it should be pointed out that the basic and the largest group of risk factors includes the risk of macroenvironment indicated by the surveyed enterprises as dominant. Therefore, the relationship between the type of risk and digital technology shows the greatest correlation strength (high or very high) between the macroenvironment risk and the selected technologies, showing the greatest effect of their implementation (high and very high) from the perspective of logistics management. This implies the claim that, depending on the digital technology, the effect of its application will be different, but the biggest impact on the logistic management process in terms of macroenvironment risk is associated with technologies such as 3D printing, artificial intelligence, blockchain, drones, augmented reality, and self-propelled vehicles. At the same time, it should be assumed that the impact and the largest set of variables will imply maintaining or increasing the competitive advantage of the enterprise (as a long-term effect) or will trigger short-term changes solved on an ad hoc basis. It should be acknowledged that the auxiliary hypothesis has been partially verified as positively here.

The above discussion also leads to certain reservations about the nature of knowledge development in the use of digital innovations in logistics management, but also in business management. Despite the relatively widespread use of digital technologies by Polish enterprises, there was no significant increase in the change in the demand (product) curve, nor was there any indication of a significant change in dynamics in the area of investment growth and return on the investment ratio. Nevertheless, this universality in the application of digital technologies determines further directions of research concerning the problem of their efficiency and beneficial pro-development changes. It should be recognized that the majority of surveyed enterprises declared that the use of digital technologies has a

regional impact, so it probably contributes only to changes in the close partnership. This is an erroneous conclusion from the point of view of the economics of the enterprise, because the implementation of digital innovations by Polish enterprises, especially in the area of logistics management, is basically at the initial stage, representing only the beginning of the transition of Polish enterprises into innovative business models. What will be important in the future is research on the possibilities of transforming digital innovations into digital business models for Polish enterprises, and the efficiency of their use in the logistics and supply chain from the points of view of efficiency and effectiveness of decision-making at the managerial level. This again leads to the claim that as the level of implementation and use of digital technologies increases, the way of risk management implicating the creation of strategic decisions will change.

## 7. Conclusions

The aim of the present study was to show the relationship between the risk of implementing digital technologies and logistics management. The literature review has proven that this issue still has a research gap in terms of management theory, including in the area of logistics. Research and analysis of the research results has shown that the implementation and use of digital technologies implies changes in logistics management. However, this impact is diversified, depending on the type of digital technology. It can therefore be assumed that not all digital technologies arouse equal interest among logistics specialists, bringing with them the yet unknown effects of the risk of implementing digital technologies. Therefore, it is justified to continue research into the type of risk of implementing digital technologies in the area of logistics management.

**Author Contributions:** Conceptualization, Ł.M. and I.D.; Methodology, A.B.; Validation, I.D., Ł.M. and A.B.; Formal analysis, A.B.; Resources, Ł.M., I.D.; Data curation, Ł.M.

**Funding:** This research received no external funding

**Conflicts of Interest:** The authors declare no conflict of interest.

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
