# Peer review of "Analysis of the Risk Impact of Implementing Digital Innovations for Logistics Management"

_processes, doi:10.3390/pr7110815_

Round 1

Reviewer 1 Report

The paper begins with a very interesting premise, to understand the risk associated with digital technologies in logistics.  The first part of the paper begins with a very general discussion of what is meant by innovation, but then fails to really develop the themes of digital innovation, and seems to throw multiple technologies (3D printing, e-commerce, driverless vehicles, and everything else) into the same bucket.  Each of these is at a very different point of adoption and deployment, so assessing risk is a function of experience, and there may be little experience with each of these technologies.

The hypotheses are introduced, but these too are very vague.  For instance:

Main hypothesis (H): The risk of introducing digital technologies in the enterprise determines changes in  logistics management.

What is meant by "change"?  This is too vague, and cannot be measured.  What is the unit of measure of change?

There is no discussion of the sample, and the description of the methodology is unclear.  How many people were contacted?  Who were they?  What are characteristics of the sample?  Without these details, the study cannot be replicated.  What is the database?  The executive summary notes that "micro, small, medium and large enterprises from the production, services, production 22 and service sectors, having a logistics unit or a logistics division, located in the “Bisnode Poland” 23 database", but little is said about this sample in the body of the paper.  In the end, the presentation of the results are also highly confusing.

The authors clearly went to a lot of work on this paper, and it deserves to be published, but not in its current form.  A major rewrite is required.

Author Response

Dear reviewer,

Thank you very much for the review. All comments are important to us and we tried to include everything as much as we could. We have made changes in the article's text.

We also send a response to your comments

Reviewer 2 Report

         I.            Line: 31-33

Being innovative is nowadays not only a challenge for enterprises, but also more and more a duty. Innovativeness is determined by the determinants of competitiveness, but also business partners and clients expect innovation.“

1.       You have to specify the determinants of competitiveness.

2.       Avoid repeating „but also“.

       II.            Line: 38-39

 „You can take a few basic ways to increase this efficiency.“

1.       Avoid using personal pronouns („you“).

2.       It would be more accurate to write „Impact of innovations on organization efficiency“, instead of „few basic ways to increase this efficiency“

3.       For easier reading, it may be better to add bullets or to rephrase (Line 42: They can also shift the company's cost curve. This aspect as the third way to achieve the goal of implementing…)

     III.            Line: 49-50

„an important and desirable aspect of enterprise innovation is the ability and knowledge of how to use digital innovation.“

You wrote that the aspect is the ability and knowledge. Is this really an aspect or something else? For example (my suggestion) „Investing in knowledge in order to use digital innovation properly/ to be able to take full advantage of digital innovations” may be considered as success factor. In this way, you will avoid misuse of the word „Aspect“. You must, of course, corroborate it by reference.

    IV.            Line: 58-59

It can be assumed that the starting point should be a  general understanding of digitization.

Digitization or digitalization?

      V.            Line: 67-68

„…or relationships into ones that are mediated by digitized artifacts and relationships.“

Digitized or digitalized artifacts? What is the difference between digitization and digitalization?

You must bear in mind that the terms are not synonymous and the difference should be emphasized.

    VI.            Line: 149-151

Table 1

You add as a source “own study” in the Literature review. What is the basis of this study, what are the references? The appropriate table would include used databases and retrieved number of paper per keywords to ensure that all the existing literature has been analyzed. The other solution is to add a table which includes main findings from various authors, but it should be corroborate by references as well.

   VII.            Line: 152-156

The implementation of digital technologies in the logistics area has led to the emergence of 152 so-called intelligent logistics (smart Logistics) [24]. M. Weiser [25] notes that the term "intelligent" 153 expresses current technological changes, which shows its dependence on time. As he argues, the 154 "smart home" of 1935 had electric light in every room, the "smart house" of 1955 had a TV set and a 155 telephone in every room, and the "smart home" of 2005 had a computer in every room.

This example is irrelevant for your topic.

 VIII.            Line: 176-179

„…efficiency of using digital technologies. Therefore, it can be concluded that research into the risk of implementing digital technology in the area of logistics management is a research gap both in the area of management and in the narrower perspective - in logistics. Therefore, the research presented in the further part of the study is justified and will fill this gap to some extent.“

You have written about research gap in the Literature review. Literature review should among others include definitions (the definitions from various authors, which are in the Introduction now, should be moved to the Literature review).

Introduction should include: definition of problem, research goal, research questions, research gap,  expected results etc.  In the introduction, the definitions should not be analyzed in detail. The introduction should introduce readers into your research.

Line: 194-196

Standardized questionnaire interviews are derived from the neo-positivist research paradigm, although their role was also played by the paradigm of the interpretive and critical postmodern paradigm.

You have to explain “neo-positivist research paradigm”, “the paradigm of the interpretive” and “critical postmodern paradigm”.

Line: 198-203

As part of this research paradigm, one strive to discover the truth about the world in a systematic, standardized, factual, synthesizing, non-subjective and cumulative way [31]. This is a well-established research method, its historical origins are combined with research conducted by  Arthur Bowley and William Benett-Hurst in Great Britain in 1912 regarding the living conditions of the working class in the cities of Stanley and Reading. However, the most significant historically is  the contribution of George Gallup, who in 1940 in the Population Census (1940 Population Census) carried out research on a five-percent sample of the American population [32].

What part did you use exactly from the research on the living conditions of the working class in the cities or on a five-percent sample of the American population? Why is this method relevant to your study about the risk impact of implementing digital innovations on logistics management?

Line: 232-233

Despite the fact that the sample was statistically correct, we analyze data from 120 companies and  not from 360.

Why did not you analyze data from 360 companies? Explain the reason.

     IX.            Line: 304

Table 3 should be translated into English.

Author Response

 Dear Reviewer,

Thank you very much for the review. All comments are important to us and we tried to include everything as much as we could. We have made changes in the article's text.

We also send a response to your comments.

Round 2

Reviewer 1 Report

The paper still has significant problems, the primary one being that I cannot understand exactly what the authors are examining! For instance:

The predefined types of dominant risk related to the implementation of digital technology 504 and the determination of the impact of risk related to the implementation of digital technology on 505 logistic management, defined by the expected effect, realizes the objective of the study set in the 506 article.

What does this mean exactly?

The authors have not done a significant edit of the paper, which needs to be done. They also need to carefully define what it is they are testing. The long table of correlation coefficients is not useful, as correlation is not equivalent to causation, which they are trying to argue that these are related to risk…

Reviewer 2 Report

"Being innovative is nowadays not only a challenge for enterprises, but also more and more a duty. Innovation is imposed on the one hand by the conditions of competition; on the other hand, its business partners and customers expect it."

Please replace "duty" with "must".

Please replace "its" with "enterprises'".

"...an important and desirable aspect of enterprise innovation is the possibility and skill to use digital innovation."

Please replace "possibility" with "ability".

From the previous review: You have to elaborate “neo-positivist research paradigm”, “the paradigm of the interpretive” and “critical postmodern paradigm”.

Author Response

Dear Reviewer,

Thank you very much for your comments. We have taken into account all comments. We have made changes in the text. We also send short answers below:

1. "Being innovative is nowadays not only a challenge for enterprises, but also more and more         a duty. Innovation is imposed on the one hand by the conditions of competition; on the other     hand, its business partners and customers expect it."

        Please replace "duty" with "must".

        Please replace "its" with "enterprises'".

        Changed as suggested

2. "...an important and desirable aspect of enterprise innovation is the possibility and skill to use         digital innovation."

        Please replace "possibility" with "ability".

        Changed as suggested

3. From the previous review: You have to elaborate “neo-positivist research paradigm”, “the                  paradigm of the interpretive” and “critical postmodern paradigm”.

    Changed as suggested. We tried to keep the clarity of expression. Explanation of these terms is     not a mainstream consideration, therefore we decided to develop these terms (as an explanation     of the meaning) in the footnotes.
